# Efficient Certified Training and Robustness Verification of Neural ODEs

**Mustafa Zeqiri, Mark Niklas Müller, Marc Fischer & Martin Vechev**
Department of Computer Science
ETH Zurich, Switzerland
mzeqiri@ethz.ch, {mark.mueller,mark.fischer,martin.vechev}@inf.ethz.ch

## Abstract

Neural Ordinary Differential Equations (NODEs) are a novel neural architecture, built around initial value problems with learned dynamics which are solved during inference. Thought to be inherently more robust against adversarial perturbations, they were recently shown to be vulnerable to strong adversarial attacks, highlighting the need for formal guarantees. However, despite significant progress in robustness verification for standard feed-forward architectures, the verification of high dimensional NODEs remains an open problem. In this work, we address this challenge and propose GAINS, an analysis framework for NODEs combining three key ideas: (i) a novel class of ODE solvers, based on variable but discrete time steps, (ii) an efficient graph representation of solver trajectories, and (iii) a novel abstraction algorithm operating on this graph representation. Together, these advances enable the efficient analysis and certified training of high-dimensional NODEs, by reducing the runtime from an intractable $\mathcal{O}(\exp(d) + \exp(T))$ to $\mathcal{O}(d + T^2 \log^2 T)$ in the dimensionality $d$ and integration time $T$. In an extensive evaluation on computer vision (MNIST and FMNIST) and time-series forecasting (Physio-Net) problems, we demonstrate the effectiveness of both our certified training and verification methods.

## 1 Introduction

As deep learning-enabled systems are increasingly deployed in safety-critical domains, developing neural architectures and specialized training methods that increase their robustness against adversarial examples (Szegedy et al., 2014; Biggio et al., 2013) – imperceptible input perturbations, causing model failures – is more important than ever. As standard neural networks suffer from severely reduced accuracies when trained for robustness, novel architectures with inherent robustness properties have recently received increasing attention (Winston & Kolter, 2020; Müller et al., 2021).

**Neural Ordinary Differential Equations** One particularly interesting such architecture are neural ODEs (NODEs) (Chen et al., 2018). Built around solving initial value problems with learned dynamics, they are uniquely suited to time-series-based problems (Rubanova et al., 2019; Brouwer et al., 2019) but have also been successfully applied to image classification (Chen et al., 2018). More importantly, NODEs have been observed to exhibit inherent robustness properties against adversarial attacks (Yan et al., 2020; Kang et al., 2021; Rodriguez et al., 2022; Zakwan et al., 2022). However, recently Huang et al. (2020) found that this robustness is greatly diminished against stronger attacks. They suggest that adaptive ODE solvers, used to solve the underlying initial value problems, cause gradient obfuscation (Athalye et al., 2018), which, in turn, causes weaker adversarial attacks to fail. This highlights the need for formal robustness guarantees to rigorously evaluate the true robustness of a model or architecture.

**Robustness Verification** For standard neural networks, many robustness verification methods have been proposed (Katz et al., 2017; Tjeng et al., 2019; Singh et al., 2018b; Raghunathan et al., 2018; Wang et al., 2021; Ferrari et al., 2022). One particularly successful class of such methods (Gehr et al., 2018; Singh et al., 2019a; Zhang et al., 2018) propagates convex shapes through the neural network that capture the reachable sets of every neuron's values and uses them to check

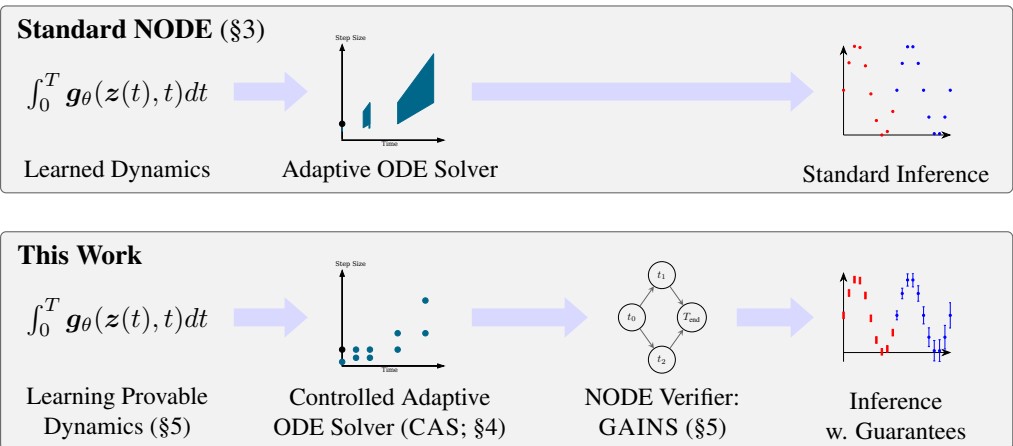

Figure 1: Overview of NODE inference in both the standard and certified (our) setting. In both settings ODE solvers are used to evaluate learned dynamics. We introduce controlled adaptive solvers (CAS) as a modification of adaptive solvers, which reduce the reachable time/step-size pairs from a continuous area to discrete points. This enables GAINS to compute worst-case bounds on NODE outputs given small input ranges, allowing both inference with guarantees and provable training. For example, in the time-series forecasting setting shown on the right, standard NODE inference predicts the blue points given the concrete red inputs. In contrast GAINS computes all possible outputs (blue error bars), for inputs in the red input ranges.

whether a given robustness property holds. Unfortunately, none of these methods can be applied to NODEs as the underlying adaptive solvers yield a continuous range of possible step-sizes (illustrated in the top panel of Fig. 1), which existing analysis techniques can not handle. First works towards NODE verification (Lopez et al., 2022) have avoided this issue by disregarding the solver behavior and analyzing only the underlying NODE dynamics in extremely low dimensional settings. However, both scaling to high-dimensional NODE architectures and taking the effect of ODE solvers into account remain open problems preventing NODE robustness verification.

**This Work** We tackle both of these problems, thereby enabling the systematic verification and study of NODE robustness (illustrated in Fig. 1) as follows: (i) We introduce a novel class of ODE solvers, based on the key insight that we can restrict step-sizes to an exponentially spaced grid with minimal impact on solver efficiency, while obtaining a finite number of time/step-size trajectories from the initial to final state (see the second column in Fig. 1). We call these **C**ontrolled **A**daptive ODE **S**olvers (CAS). Unfortunately, CAS solvers still yield exponentially many trajectories in the integration time. (ii) We, therefore, introduce an efficient graph representation, allowing trajectories to be merged, reducing their number to quadratically many. (iii) We develop a novel algorithm for the popular DEEPPOLY convex relaxation (Singh et al., 2019a), effective for standard neural network verification, that enables the efficient application of DEEPPOLY to the trajectory graph by handling trajectory splitting in linear instead of exponential time. Combining these core ideas, we propose GAINS[1], a novel framework for certified training and verification of NODEs that leverages key algorithmic advances to achieve polynomial runtimes and allows us to faithfully assess the robustness of NODEs.

**Main Contributions** Our main contributions are:

- A novel class of ODE solvers, CAS solvers, retaining the efficiency of adaptive step size solvers while enabling verification (§4).
- An efficient linear bound propagation based framework, GAINS, which leverages CAS to enable the efficient training and verification of NODEs (§5).
- An extensive empirical evaluation demonstrating the effectiveness of our method in ablation studies and on image classification and time-series problems (§6).

---

[1]**G**raph based **A**bstract **I**nterpretation for **NODE**s

## 2 ADVERSARIAL ROBUSTNESS

In this section, we discuss the necessary background relating to adversarial robustness.

**Adversarial Robustness** We consider both classification and regression models $\boldsymbol{f_\theta} \colon \mathbb{R}^{d_{\mathrm{in}}} \mapsto \mathbb{R}^c$ with parameters $\boldsymbol{\theta}$ that, given an input $\boldsymbol{x} \in \mathcal{X} \subseteq \mathbb{R}^{d_{\mathrm{in}}}$, predict $c$ numerical values $\boldsymbol{y} := \boldsymbol{f}(\boldsymbol{x})$, interpreted as class confidences or predictions of the regression values, respectively. In the classification setting, we call $\boldsymbol{f}$ adversarially robust on an $\ell_p$-norm ball $\mathcal{B}_p^{\epsilon_p}(\boldsymbol{x})$ of radius $\epsilon_p$, if it predicts target class $t$ for all perturbed inputs $\boldsymbol{x}' \in \mathcal{B}_p^{\epsilon_p}(\boldsymbol{x})$. More formally, we define *adversarial robustness* as:

$$\arg\max_j h(\boldsymbol{x}')_j = t, \quad \forall \boldsymbol{x}' \in \mathcal{B}_p^{\epsilon_p}(\boldsymbol{x}) := \{x' \in \mathcal{X} \mid \|\boldsymbol{x} - \boldsymbol{x}'\|_p \le \epsilon_p\}. \tag{1}$$

In the regression setting, we evaluate $\nu$-$\delta$-robustness by checking whether the worst-case mean absolute error $\mathrm{MAE}_{\mathrm{rob}}(\boldsymbol{x})$ for $\boldsymbol{x}' \in \mathcal{B}_p^{\epsilon_p}(\boldsymbol{x})$ is linearly bounded by the original input's $\mathrm{MAE}(\boldsymbol{x})$:

$$\mathrm{MAE}_{\mathrm{rob}}(\boldsymbol{x}) < (1+\nu)\,\mathrm{MAE}(\boldsymbol{x}) + \delta, \quad \text{with} \quad \mathrm{MAE}_{\mathrm{rob}}(\boldsymbol{x}) = \max_{\boldsymbol{x}' \in \mathcal{B}_p^{\epsilon_p}(\boldsymbol{x})} \mathrm{MAE}(\boldsymbol{x}'). \tag{2}$$

**Adversarial Attacks** aim to disprove robustness properties by finding a concrete counterexample $\boldsymbol{x}'$. A particularly successful such method is the PGD attack (Madry et al., 2018), which computes $\boldsymbol{x}'$ by initializing $\boldsymbol{x}'_0$ uniformly at random in $\mathcal{B}_p^{\epsilon_p}(\boldsymbol{x})$ and then updating it in the direction of the gradient sign of an auxiliary loss function $\mathcal{L}$, using $N$ projected gradient descent steps:

$$\boldsymbol{x}'_{n+1} = \Pi_{\mathcal{B}_p^{\epsilon_p}(\boldsymbol{x})} \boldsymbol{x}'_n + \alpha \operatorname{sign}(\nabla_{\boldsymbol{x}'_n} \mathcal{L}(\boldsymbol{f_\theta}(\boldsymbol{x}'_n), t)), \tag{3}$$

where $\Pi_S$ denotes projection on $S$ and $\alpha$ the step size. We say an input $\boldsymbol{x}$ is empirically robust if no counterexample $\boldsymbol{x}'$ is found.

**Neural Network Verification** aims to decide whether the robustness properties defined above hold. To this end, a wide range of methods has been proposed, many relying on bound propagation, i.e., determining a lower and upper bound for each neuron $l \le x \le u$, or in vector notation for the whole layer $\boldsymbol{l} \le \boldsymbol{x} \le \boldsymbol{u}$. Here, we discuss two ways of obtaining such bounds: First, *Interval Bound Propa-*

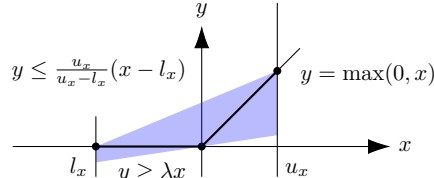

Figure 2: Linear bounds for $\mathrm{ReLU}(x)$.

*gation* (Gehr et al., 2018; Mirman et al., 2018) where $\boldsymbol{l}$ and $\boldsymbol{u}$ are constants, bounding the reachable values of neurons. For details, we refer to Gowal et al. (2018). Second, *Linear Bound Propagation* (Singh et al., 2019a; Zhang et al., 2018), where every layer's neurons $\boldsymbol{x}_i$ are lower- and upper-bounded depending only on the previous layer's neurons:

$$\boldsymbol{A}_i^- \boldsymbol{x}_{i-1} + \boldsymbol{c}_i^- =: \boldsymbol{l}_i \le \boldsymbol{x}_i, \quad \boldsymbol{x}_i \le \boldsymbol{u}_i := \boldsymbol{A}_i^+ \boldsymbol{x}_{i-1} + \boldsymbol{c}_i^+. \tag{4}$$

Given these linear constraints, we can recursively substitute $\boldsymbol{x}_{i-1}$ with its linear bounds in terms of $\boldsymbol{x}_{i-2}$ until we have obtained bounds depending only on the input $\boldsymbol{x}_0$. This allows us to compute concrete bounds $\boldsymbol{l}$ and $\boldsymbol{u}$ on any linear expression over network neurons.

For a linear layer $\boldsymbol{x}_i = \boldsymbol{W}_i \boldsymbol{x}_{i-1} + \boldsymbol{b}_i$ we simply have $\boldsymbol{A}_i^\pm = \boldsymbol{W}_i$ and $\boldsymbol{c}_i^\pm = \boldsymbol{b}_i$. For a ReLU function $\boldsymbol{x}_i = \mathrm{ReLU}(\boldsymbol{x}_{i-1})$, we first compute the input bounds $\boldsymbol{l} \le \boldsymbol{x}_{i-1} \le \boldsymbol{u}$. If the ReLU is stably inactive, i.e. $u \le 0$, we can replace it with the zero-function. If the ReLU is stably active, i.e. $l \ge 0$, we can replace it with the identity-function. In both cases, we can use the bounding for a linear layer. If the ReLU is unstable, i.e., $l < 0 < u$, we compute a convex relaxation with parameter $\lambda$ as illustrated in Fig. 2. Using this backsubstitution approach, we can now lower bound the difference $y_t - y_i$, $\forall i \ne t$ to determine whether the target class logit $y_t$ is always greater than all other logits in the classification setting and similarly bound the elementwise output range in the regression setting.

**Provable Training** Special training is necessary to obtain networks that are provably robust. Considering the classification setting with a data distribution $(\boldsymbol{x}, t) \sim \mathcal{D}$. Provable training now aims to choose the network parametrization $\boldsymbol{\theta}$ that minimizes the expected *worst case* loss:

$$\boldsymbol{\theta}_{\mathrm{rob}} = \arg\min_{\boldsymbol{\theta}} \mathbb{E}_{\mathcal{D}} \Big[ \max_{\boldsymbol{x}' \in \mathcal{B}_p^{\epsilon_p}(\boldsymbol{x})} \mathcal{L}_{\mathrm{CE}}(\boldsymbol{f_\theta}(\boldsymbol{x}'), t) \Big] \quad \text{with} \quad \mathcal{L}_{\mathrm{CE}}(\boldsymbol{y}, t) = \ln\big(1 + \sum_{i \ne t} \exp(y_i - y_t)\big). \tag{5}$$

The inner maximization problem is generally intractable, but can be upper bounded using bound propagation (Mirman et al., 2018; Gowal et al., 2018; Zhang et al., 2020; Müller et al., 2023).

## 3 NEURAL ORDINARY DIFFERENTIAL EQUATIONS

In this section, we discuss the necessary background relating to NODEs (Chen et al., 2018).

**Neural Ordinary Differential Equations** are built around an initial value problem (IVP), defined by an input state $z(0) = z_0$ and a neural network $g_\theta$ defining the dynamics of an ordinary differential equation (ODE) $\nabla_t z(t) = g_\theta(z(t), t)$. We obtain its solution $z(T)$ at time $T$ as

$$z(T) = z(0) + \int_0^T g_\theta(z(t), t)dt. \tag{6}$$

Generally, $z_0$ can either be the raw input or come from an encoder neural network. For both classification and regression tasks, we output $y = f_\theta(z(T_{\text{end}}))$ for an input $z_0$ and a predefined $T_{end}$, where $f$ is an additional decoder, usually a linear layer.

Time series forecasting is a special case of the regression setting where the input is a time-series $x_{ts}^L = \{(x_j, t_j)\}_{j=1}^L$, defined as a sequence of $L$ entries, each consisting of a data point $x_j \in \mathbb{R}^{d_{\text{in}}}$ and an observation time $t_j$. We aim to predict the value of the last observed data point $x_L$, using only the first $L' < L$ data points as input. To this end, we employ the so-called latent-ODE architecture, where a recurrent encoder network reads the data sequence $\{(x_j, t_j)\}_{j=1}^{L'}$ and outputs the initial state $z_0$ for a decoder NODE that is then integrated up to the desired time-step $T_{\text{end}} = t_L$ before its output $z_{t_L}$ is passed through a linear layer $f$. For further details, we refer to App. A.

**ODE Solvers** are employed to approximate Eq. (6), as analytical solutions often don't exist. These solvers split the integration interval $[0, T]$ into sub-intervals, for which the integral is numerically approximated by evaluating $g_\theta$ at multiple points and taking their weighted average. We let $\Gamma(z_0)$ denote the trajectory of an ODE solver, which we define as the sequence of tuples $(t, h)$ with time $t$ and step-size $h$.

ODE solvers are characterized by their order $p$, indicating how quickly approximation errors diminish as the step size is reduced (Shampine, 2005). We distinguish between fixed ($h$ constant) (Euler, 1792; Runge, 1895) and adaptive solvers ($h$ varies; discussed below) (Dormand & Prince, 1980; Bogacki & Shampine, 1989). Note that for adaptive solvers, the trajectory depends on the exact input. Huang et al. (2020) found that the supposedly inherent robustness of NODEs to adversarial attacks (Kang et al., 2021; Yan et al., 2020) is only observed for adaptive ODE solvers and may stem, partially or entirely, from gradient obfuscation (Athalye et al., 2018) caused by the solver.

**Adaptive ODE Solvers** Adaptive step-size solvers (Dormand & Prince, 1980; Bogacki & Shampine, 1989) use two methods of different order to compute the proposal solutions $\hat{z}^1(t + h)$ and $\hat{z}^2(t + h)$ and derive an error estimate $\delta = \|\frac{\hat{z}^1(t+h) - \hat{z}^2(t+h)}{\tau}\|_1$, normalized by the absolute error tolerance $\tau$. This error estimate $\delta$ is then used to update the step size $h \leftarrow h\delta^{-1/p}$. Next, we discuss the challenges this poses for robustness verification and how we tackle them.

## 4 CONTROLLED ADAPTIVE ODE SOLVERS

Adaptive ODE solvers (AS) update their step-size $h$ continuously depending on the error estimate $\delta$. For continuous input regions, this generally yields infinitely many trajectories, making their abstraction intractable. We illustrate this in Fig. 3 (details in App. C.2), where the blue regions (■) mark all (time, step-size) tuples that are reachable after two steps. To overcome this, we propose controlled adaptive solvers (CAS), which restrict step-sizes to a discrete set (●), making them amenable to certification (§5). Next, we show how any adaptive ODE solver can be converted into a corresponding CAS solver.

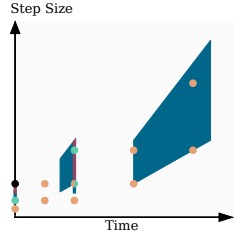

Figure 3: We compare CAS and adaptive solvers (AS) with respect to the reachable time/step-size tuples after one (●, ■) and two solver steps (●, ■).

**Step-Size Update** We modify the step-size update rule of any AS as

$$h \leftarrow \begin{cases} h \cdot \alpha, & \text{if } \delta \leq \tau_\alpha, \\ h, & \text{if } \tau_\alpha < \delta \leq 1, \\ h/\alpha, & \text{otherwise.} \end{cases} \qquad \delta = \left\| \frac{\hat{z}^1(t + h) - \hat{z}^2(t + h)}{\tau} \right\|_1,$$

with update factor $\alpha \in \mathbb{N}^{>1}$, and the $\alpha$-induced decision threshold $\tau_\alpha = \alpha^{-p}$. Intuitively, we increase the step size by a factor $\alpha$ if we expect the normalized error after this increase to still be acceptable, i.e., $\delta \leq \alpha^{-p}$, we decrease the step size by a factor $\alpha$ and repeat the step if the error exceeds our tolerance, i.e., $\delta > 1$, and we keep the same step size otherwise. If the time $t + h$ after the next step would exceed the final time $T_{\text{end}}$, we clip the step size to $h \leftarrow \min(h, T_{end} - t)$. Additionally, we enforce a minimum step-size. For more details, see App. C.1.

We contrast the update behaviors of CAS and AS solvers in Fig. 3. We initialize both solvers with the same state (●) and after one step, the CAS solver can reach exactly three different states (●) while the adaptive solver can already reach continuous states (■). After two steps this difference becomes even more clear with the CAS solver reaching only 9 states (●) while the adaptive solver can reach a large region of time/step-size combinations (■).

**Initial Step-Size**    During training, the initial step size $h_0$ is computed based on the initial state and corresponding gradient. To avoid this dependence during inference, we always use its exponentially weighted average, computed during training (details in App. C.1).

**Comparison to Adaptive Solvers**    CAS solvers can be seen as adaptive solvers with discretized step-sizes of the same order. Due to the exponentially spaced step-sizes, CAS can approximate any step-size chosen by an AS up to a factor of at most $\alpha$, with the CAS always choosing the smaller steps. Thus, CAS will need at most $\alpha$-times as many steps as an adaptive solver, assuming that the adaptive solver will never update the step size by more than $\alpha$ in one step. Empirically, we confirm this on a conventional non-linear ODE, plotting mean absolute errors over the mean number of solver steps depending on the error threshold in Fig. 4. There, we see that a dopri5-based CAS solver performs similarly to an unmodified dopri5 (AS). For more details and additional comparisons between the solvers, we refer to App. C.2 and App. H.1.

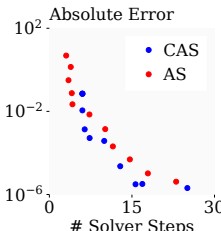

Figure 4: AS and CAS error over solver steps.

## 5    VERIFICATION OF NEURAL ORDINARY DIFFERENTIAL EQUATIONS

While the discrete step sizes of CAS, discussed in §4, yield a finite number of trajectories for any input region, there are still exponentially many in the integration time. Naively computing bounds for all of them independently is thus still intractable. To tackle this challenge, we introduce the analysis framework GAINS, short for **G**raph based **A**bstract **I**nterpretation for **NODE**s, which allows us to efficiently propagate bounds through the ODE solver using a graph representation of all trajectories. We discuss two instantiations, one using interval bounds, the other linear bounds.

Let us consider a NODE with input $\mathcal{Z}$, either obtained from an encoder or directly from the data. We now define the trajectory graph $\mathcal{G}(\mathcal{Z}) = (\mathcal{V}, \mathcal{E})$, representing all trajectories $\Gamma(z'_0)$ for $z'_0 \in \mathcal{Z}$. The nodes $v \in \mathcal{V}$ represent solver states $(t, h)$ with time $t$ and step-size $h$ and aggregate interval bounds on the corresponding $z(t)$. The directed edges $e \in \mathcal{E}$ connect consecutive states in possible solver trajectories. This representation allows us to merge states $z(t)$ with identical time and step-size, regardless of the trajectory taken to reach them. This reduces the number of trajectories or rather solver steps we have to consider from exponential $\mathcal{O}(\exp(T_{\text{end}}))$ to quadratic $\mathcal{O}(T_{\text{end}}^2 \log^2(T_{\text{end}}))$ (given at most $\mathcal{O}(T_{\text{end}} \log(T_{\text{end}}))$ nodes in $\mathcal{V}$ as derived in App. B), making the analysis tractable.

**Verification with Interval Bounds**    We first note that each solver step only consists of computing the weighted sum of evaluations of the network $g$, allowing standard interval bound propagation to be used for its abstraction. We call this evaluation of a solver on a set of inputs an *abstract solver step*. Now, given an input $\mathcal{Z}$, we construct our trajectory graph $\mathcal{G}(\mathcal{Z})$ as follows: We do an abstract solver step, compute the interval bounds of the local error estimate $\delta_{(t,h)}$, and check which step size updates (increase, accept, or decrease) could be made according to the CAS. Depending on the looseness of the bounds, multiple updates might be chosen; we call this case trajectory splitting. For each possible update, we obtain a new state tuple $(t', h')$ and add the node $(t', h')$ to $\mathcal{V}$ and an edge from $(t, h)$ to $(t', h')$ to $\mathcal{E}$. If the node $(t', h')$ already existed, we update its state to contain the convex hull of the interval bounds. We repeat this procedure until all trajectories have reached the termination node $(T_{\text{end}}, 0)$. This yields a complete trajectory graph and interval bounds for $z(T_{\text{end}})$.

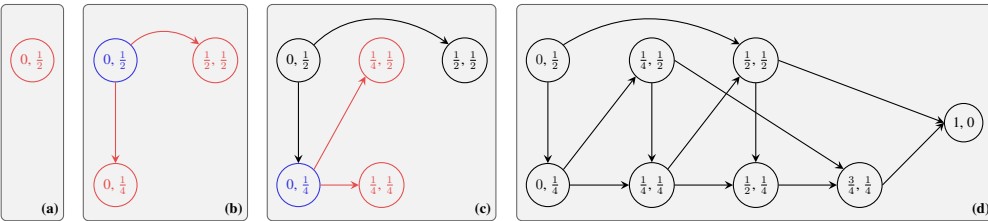

Figure 5: An example trajectory graph $\mathcal{G}(\mathcal{Z})$ construction for a controlled adaptive ODE solver with $h_0 = \frac{1}{2}$, $\alpha = 2$ and $T_{end} = 1$. Note how trajectory splitting occurs in all vertices except the last two states.

If there are further layers after the NODE, standard interval propagation can be employed to obtain the network output $\boldsymbol{y}$.

We illustrate this construction process in Fig. 5, where we highlight newly added edges and nodes in red and the processed node in blue: We initialize the graph with the node $(0, h_0)$, in our case $h_0 = 1/2$ (see Fig. 5(a)). We now do an abstract solver step for this node and find that $\delta > \tau_\alpha$. Thus, we either accept the step, yielding the next node $(1/2, 1/2)$, or we reject the step and decrease the step-size by $\alpha = 2$, yielding the node $(0, 1/4)$, both are connected to the current node (see Fig. 5 (b)). We now choose among the nodes without outgoing edges the one with the smallest current time $t$ and largest step-size $h$ (in that order), $(0, 1/4)$ in our case, and do another abstract solver step, yielding $\delta < 1$. We thus either accept the step, yielding the node $(1/4, 1/4)$, or additionally increase the step-size, yielding the node $(1/4, 1/2)$ (see Fig. 5 (c)). We proceed this way until the only node without outgoing edges is the termination node $(T_{end}, 0)$ with $T_{end} = 1$ in our case (see Fig. 5 (d)).

**Verification with Linear Bounds** To compute more precise linear bounds on $\boldsymbol{z}(T_{end})$, we first construct the trajectory graph $\mathcal{G}(\mathcal{Z})$ as discussed above, using either interval bounds or the linear bounding procedure described below, retaining concrete element-wise upper and lower bounds at every state. We can now derive linear bounds on $\boldsymbol{z}(T_{end})$ in terms of the NODE input $\boldsymbol{z}_0$ by recursively substituting bounds from intermediate computation steps. Starting with the bounds for $(T_{end}, 0)$, we backsubstitute them along every incoming edge, yielding a set of bounds in every preceding node. We recursively repeat this procedure until we arrive at the input node. We illustrate this in Fig. 6, where we, starting at $T_{end}$, backsubstitute $\boldsymbol{y}$ to $t_1$ and $t_2$, obtaining bounds in terms of $\boldsymbol{z}_1$ and $\boldsymbol{z}_2$. In contrast to the standard DEEPPOLY backward substitution procedure, a node

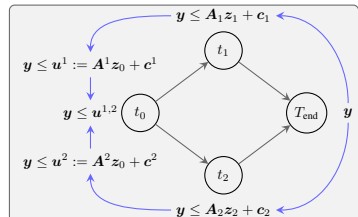

Figure 6: Example upper bounds for $\boldsymbol{y} = \boldsymbol{z}(T_{end})$ via GAINS. (Lower bounds analogous.) Blue arrows show the backward substitution resulting in LCAP at $t_0$.

in $\mathcal{G}(\mathcal{Z})$ can have multiple successors which reach the final node via different trajectories. We can thus obtain several sets of linear constraints bounding the same expression with respect to the same state, which we need to merge in a sound manner without losing too much precision. We call this the linear constraint aggregation problem (LCAP) and observe that it arises in Fig. 6 after an additional backsubstitution step to $t_0$ yields two bounds, $\boldsymbol{u}^1$ and $\boldsymbol{u}^2$, on $\boldsymbol{y}$ both in terms of $\boldsymbol{z}_0$.

**Linear Constraint Aggregation Problem** The LCAP requires us to soundly merge a set of different linear constraints bounding the same variable. As an example, we consider a variable $\boldsymbol{y} = \boldsymbol{z}(T)$ for which we have $m$ upper bounds $\{\boldsymbol{u}^j\}_{j=1}^m$ linear in $\boldsymbol{z}_0$, which in turn can take values in $\mathcal{Z}$. In this case, we want to obtain a single linear upper bound $\boldsymbol{y} \leq \boldsymbol{a}\boldsymbol{z}_0 + c$ that minimizes the volume between the constraint and the $y = 0$ plane over $\boldsymbol{z}_0 \in \mathcal{Z}$, while soundly over-approximating all constraints. More formally, we want to solve:

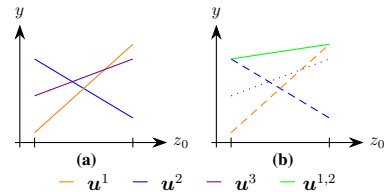

Figure 7: Visualization of the LCAP with $m = 3$, shown in (a). In (b) the constraints $\boldsymbol{u}^1, \boldsymbol{u}^2$ (dashed) are over-approximated by $\boldsymbol{u}^{1,2}$ via CURLS, which also bounds $\boldsymbol{u}^3$ (dotted).

$$\underset{\boldsymbol{a},c}{\arg\min} \int_{\mathcal{Z}} \boldsymbol{a}\boldsymbol{z}_0 + c \, \mathrm{d}\boldsymbol{z}_0, \quad s.t. \ \boldsymbol{a}\boldsymbol{z}_0 + c \geq \max_j \boldsymbol{a}^j \boldsymbol{z}_0 + c^j, \quad \forall \boldsymbol{z}_0 \in \mathcal{Z}. \tag{7}$$

Table 1: Means and standard deviations of the standard (Std.), adversarial (Adv.), and certified (Cert.) accuracy obtained with GAINS depending on the training method and evaluated on the first 1000 test set samples.

| Dataset | Training Method | $\epsilon_t$ | Std. [%] | $\epsilon = 0.10$ | | $\epsilon = 0.15$ | | $\epsilon = 0.20$ | |
|---|---|---|---|---|---|---|---|---|---|
| | | | | Adv. [%] | Cert. [%] | Adv. [%] | Cert. [%] | Adv. [%] | Cert. [%] |
| MNIST | Standard | | $98.8^{\pm0.4}$ | $23.2^{\pm3.5}$ | $0.0^{\pm0.0}$ | $2.5^{\pm1.6}$ | $0.0^{\pm0.0}$ | $0.3^{\pm0.2}$ | $0.0^{\pm0.0}$ |
| | Adv. | 0.11 | $\mathbf{99.2}^{\pm0.1}$ | $\mathbf{95.4}^{\pm0.4}$ | $0.0^{\pm0.0}$ | $\mathbf{88.3}^{\pm0.6}$ | $0.0^{\pm0.0}$ | $59.4^{\pm3.2}$ | $0.0^{\pm0.0}$ |
| | GAINS | 0.11 | $95.5^{\pm0.1}$ | $91.5^{\pm0.6}$ | $\mathbf{89.0}^{\pm1.1}$ | $84.0^{\pm2.7}$ | $47.2^{\pm7.9}$ | $21.4^{\pm1.8}$ | $0.2^{\pm0.2}$ |
| | | 0.22 | $91.8^{\pm1.3}$ | $88.5^{\pm1.8}$ | $86.8^{\pm2.0}$ | $86.8^{\pm2.1}$ | $\mathbf{83.7}^{\pm2.3}$ | $\mathbf{84.5}^{\pm3.2}$ | $\mathbf{79.7}^{\pm3.4}$ |
| FMNIST | Standard | | $\mathbf{88.6}^{\pm1.2}$ | $0.1^{\pm0.1}$ | $0.0^{\pm0.0}$ | $0.0^{\pm0.0}$ | $0.0^{\pm0.0}$ | | |
| | Adv. | 0.11 | $80.9^{\pm0.7}$ | $\mathbf{70.2}^{\pm0.5}$ | $0.0^{\pm0.0}$ | $47.1^{\pm3.7}$ | $0.0^{\pm0.0}$ | | |
| | GAINS | 0.11 | $75.1^{\pm1.2}$ | $65.7^{\pm1.0}$ | $\mathbf{62.5}^{\pm1.1}$ | $21.1^{\pm5.9}$ | $13.3^{\pm3.1}$ | | |
| | | 0.16 | $71.5^{\pm1.7}$ | $64.0^{\pm2.7}$ | $61.3^{\pm2.7}$ | $\mathbf{60.1}^{\pm3.5}$ | $\mathbf{55.0}^{\pm4.3}$ | | |

While this can be cast as a linear program by enumerating all exponentially many corners of $\mathcal{Z}$, this becomes intractable even in modest dimensions. To overcome this challenge, we propose **C**onstraint **U**nification via **ReLU S**implification (CURLS), translating the $\max\{\boldsymbol{u}^j\}_{j=1}^m$ into a composition of ReLUs, which can be handled using the efficient DEEPPOLY primitive proposed by Singh et al. (2019a). For a pair of constraints $\boldsymbol{u}_i^1, \boldsymbol{u}_i^2$ we can rewrite their maximum as

$$\max_{j \in 1,2} \boldsymbol{u}_i^j = \boldsymbol{u}_i^1 + \max(0, \boldsymbol{u}_i^2 - \boldsymbol{u}_i^1) = \boldsymbol{u}_i^1 + \mathrm{ReLU}(\boldsymbol{u}_i^2 - \boldsymbol{u}_i^1). \tag{8}$$

In the case of $m$ constraints, this rewrite can be applied multiple times. We note that lower bounds can be merged analogously and visualize CURLS for a 1-d problem in Fig. 7. There, the first iteration already yields the constraint $\boldsymbol{u}^{1,2}$, dominating the remaining $\boldsymbol{u}^3$.

**Training**   In order to train NODEs amenable to verification we utilize the box bounds discussed above and sample $\kappa$ trajectories form $\mathcal{G}(\mathcal{Z})$. For more details, please see App. B.

**Bound Calculation**   During the computation of the bounds, GAINS combines verification with interval and linear bounds by using the tighter bound of either approach (more details in App. C.3).

## 6 EXPERIMENTAL EVALUATION

**Experimental Setup**   We implement GAINS in PyTorch[2] (Paszke et al., 2019) and evaluate all benchmarks using single NVIDIA RTX 2080Ti. We conduct experiments on MNIST (LeCun et al., 1998), FMNIST (Xiao et al., 2017), and PHYSIO-NET (Silva et al., 2012). For image classification, we use an architecture consisting of two convolutional and one NODE layer (see Table 5 in App. D for more details). For time-series forecasting, we use a latent ODE (see Table 6 in App. E for more details). We provide detailed hyperparameter choices in App. D and E.

### 6.1 CLASSIFICATION

We train NODE based networks with standard, adversarial, and provable training ($\epsilon_t \in \{0.11, 0.22\}$) and certify robustness to $\ell_\infty$-norm bounded perturbations of radius $\epsilon$ as defined in Eq. (1). We report means and standard deviations across three runs at different perturbation levels ($\epsilon \in \{0.1, 0.15, 0.2\}$) depending on the training method in Table 1. Both for MNIST and FMNIST, adversarial accuracies are low ($0.0\%$ to $23.2\%$) for standard trained NODEs, agreeing well with recent observations showing vulnerabilities to strong attacks (Huang et al., 2020). While adversarial training can significantly improve robustness even against these stronger attacks, we can not certify any robustness. Using provable training with GAINS significantly improves certifiable accuracy (to up to $89\%$ depending on the setting) while reducing standard accuracy only moderately. This trade-off becomes more pronounced as we consider increasing perturbation magnitudes for training and certification.

### 6.2 TIME-SERIES FORECASTING

For time-series forecasting, we consider the PHYSIO-NET (Silva et al., 2012) dataset, containing $8\,000$ time-series of up to $48$ hours of $35$ irregularly sampled features. We rescale most features

---

[2]We release our code at `https://github.com/eth-sri/GAINS`

Table 2: Comparison of the mean absolute errors for the unperturbed samples (Std. MAE), and the adversarial (Adv.), and certifiable (Cert.) $\nu$-$\delta$-robustness with $\nu = 0.1$ and $\delta = 0.01$ obtained using different provable training methods on the full PHYSIO-NET test set.

| Setting | Training Method | $\epsilon_t$ | Std. MAE [$\times 10^{-2}$] | $\epsilon = 0.05$ | | $\epsilon = 0.10$ | | $\epsilon = 0.20$ | |
|---|---|---|---|---|---|---|---|---|---|
| | | | | Adv. [%] | Cert. [%] | Adv. [%] | Cert. [%] | Adv. [%] | Cert. [%] |
| 6h | Standard | | $\mathbf{47.4}^{\pm 0.3}$ | $54.3^{\pm 3.8}$ | $0.0^{\pm 0.0}$ | $13.7^{\pm 2.9}$ | $0.0^{\pm 0.0}$ | $2.3^{\pm 1.2}$ | $0.0^{\pm 0.0}$ |
| | GAINS | 0.1 | $51.1^{\pm 2.0}$ | $97.7^{\pm 0.7}$ | $93.0^{\pm 2.7}$ | $77.0^{\pm 7.3}$ | $60.4^{\pm 10.9}$ | $42.0^{\pm 11.0}$ | $24.2^{\pm 7.7}$ |
| | | 0.2 | $57.6^{\pm 2.5}$ | $\mathbf{100.0}^{\pm 0.0}$ | $\mathbf{99.8}^{\pm 0.1}$ | $\mathbf{96.4}^{\pm 2.1}$ | $\mathbf{93.1}^{\pm 4.5}$ | $\mathbf{80.1}^{\pm 11.7}$ | $\mathbf{70.5}^{\pm 18.9}$ |
| 12h | Standard | | $\mathbf{49.9}^{\pm 0.2}$ | $65.2^{\pm 2.0}$ | $0.0^{\pm 0.0}$ | $16.6^{\pm 2.3}$ | $0.0^{\pm 0.0}$ | $2.0^{\pm 0.4}$ | $0.0^{\pm 0.0}$ |
| | GAINS | 0.1 | $50.9^{\pm 0.4}$ | $98.0^{\pm 0.2}$ | $94.5^{\pm 0.7}$ | $74.3^{\pm 3.5}$ | $55.8^{\pm 1.5}$ | $28.9^{\pm 3.6}$ | $17.2^{\pm 0.1}$ |
| | | 0.2 | $52.9^{\pm 0.1}$ | $\mathbf{99.1}^{\pm 0.1}$ | $\mathbf{98.3}^{\pm 0.2}$ | $\mathbf{87.8}^{\pm 0.8}$ | $\mathbf{80.3}^{\pm 0.8}$ | $\mathbf{52.3}^{\pm 0.8}$ | $\mathbf{38.5}^{\pm 1.7}$ |
| 24h | Standard | | $\mathbf{51.2}^{\pm 0.3}$ | $69.7^{\pm 1.9}$ | $0.0^{\pm 0.0}$ | $23.6^{\pm 2.8}$ | $0.0^{\pm 0.0}$ | $3.7^{\pm 1.0}$ | $0.0^{\pm 0.0}$ |
| | GAINS | 0.1 | $51.5^{\pm 0.1}$ | $97.9^{\pm 0.2}$ | $96.2^{\pm 0.4}$ | $78.3^{\pm 2.3}$ | $68.0^{\pm 1.6}$ | $32.6^{\pm 0.6}$ | $22.7^{\pm 1.0}$ |
| | | 0.2 | $53.7^{\pm 0.7}$ | $\mathbf{99.7}^{\pm 0.1}$ | $\mathbf{99.1}^{\pm 0.3}$ | $\mathbf{92.3}^{\pm 1.7}$ | $\mathbf{89.4}^{\pm 2.4}$ | $\mathbf{59.8}^{\pm 7.7}$ | $\mathbf{50.5}^{\pm 5.1}$ |

to mean $\mu = 0$ and standard deviation $\sigma = 1$ (before applying perturbations) and refer to App. E for more details. We consider three settings, where we predict the last measurement $L$, without having access to the preceding 6, 12, or 24 hours of data. In Table 2, we report the mean absolute prediction error (MAE) for the unperturbed samples and $\nu$-$\delta$-robustness (see Eq. (2)) for relative and absolute error tolerances of $\nu = 0.1$ and $\delta = 0.01$, respectively, at perturbation magnitudes $\epsilon = \{0.05, 0.1, 0.2\}$. We observe only a minimal drop in standard precision, when certifiably training with GAINS at moderate perturbation magnitudes ($\epsilon_t = 0.1$) while increasing both adversarial and certified accuracies substantially. Interestingly, the drop in standard precision is the biggest for the $6h$ setting, despite having the shortest forecast horizon among all settings. We hypothesize that this is due to the larger number of input points and thus abstracted embedding steps leading to increased approximation errors. Further, while we can again not verify any robustness for standard trained NODEs, they exhibit non-vacuous empirical robustness. However, without guarantees it remains unclear whether this is due to adversarial examples being harder to find or NODEs being inherently more robust. Across settings, we observe that training with larger perturbation magnitudes leads to slightly worse performance on unperturbed data, but significantly improves robustness.

## 6.3 ABLATION

**Trajectory Sensitivity** We investigate whether the solver trajectory, i.e., the chosen step-sizes, of CAS solvers are susceptible to adversarial perturbations by conducting an adversarial attack aiming directly to change the trajectory $\Gamma(z_0)$ (see App. F for more details). In Table 3, we report the success rate of this attack for MNIST, showing that even at moderate perturbation magnitudes ($\epsilon = 0.1$) attacks are (almost) always successful if models are trained using standard or adversarial training. While training with

Table 3: Mean and standard deviation of the attack success [%] on the first 1000 samples of the MNIST test set.

| Training | $\epsilon_t$ | Attack Success [%] | | |
|---|---|---|---|---|
| | | $\epsilon = 0.1$ | $\epsilon = 0.15$ | $\epsilon = 0.2$ |
| Standard | | $98.9^{\pm 0.3}$ | $100.0^{\pm 0.1}$ | $100.0^{\pm 0.0}$ |
| Adversarial | 0.11 | $99.3^{\pm 0.1}$ | $100.0^{\pm 0.0}$ | $100.0^{\pm 0.0}$ |
| GAINS | 0.11 | $73.4^{\pm 3.5}$ | $86.3^{\pm 3.5}$ | $95.5^{\pm 1.8}$ |
| | 0.22 | $65.2^{\pm 7.5}$ | $75.3^{\pm 6.2}$ | $82.2^{\pm 5.0}$ |

GAINS reduces this susceptibility notably, it remains significant. This highlights the need to consider the effect of a chosen solver on robustness, motivating both the use of CAS solvers and the trajectory graph-based approach of GAINS.

**Linear Constraint Aggregation** To evaluate CURLS on the Linear Constraint Aggregation problem (LCAP), we compare it to an LP-based approach based on Eq. (7) and implemented using a commercial LP solver (GUROBI (Gurobi Optimization, LLC, 2022)). However, considering all soundness constraints associated with the $2^d$ corner points is intractable. Therefore, we use an iterative sampling strategy (see App. G for more details).

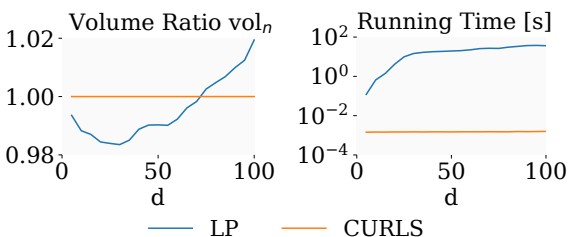

Figure 8: Comparison of the CURLS and LP solution to the LCAP with respect to normalized volume (left) and runtime (right).

To compare the obtained relational constraints, we consider the volumes induced by the two methods and report mean normalized abstraction volumes $\text{vol}^{\text{LP}} / \text{vol}^{\text{CURLS}}$ in Fig. 8 for sets of $m = 4$ randomly generated constraints in $d = [5, 100]$ dimensions (see App. G for more details). We observe that while the LP based solutions are more precise for up to 75 dimensional problems, they take around 5 orders of magnitude longer to compute. For higher dimensional problems, CURLS is both faster and more precise. During the certification of a single input, we consider multiple hundred up to 512 dimensional LCAP problems, making even the sampling based LP solution infeasible in practice and highlighting the importance of the efficient constraint aggregation via CURLS for the GAINS framework.

## 7 RELATED WORK

**Empirical Robustness of NODEs**   Yan et al. (2020) introduce TisODEs, by adding a regularization term to the loss which penalizes differences between neighboring trajectories to improve empirical robustness. A range of work (Kang et al., 2021; Rodriguez et al., 2022; Huang et al., 2020; Zakwan et al., 2022) trains NODEs which satisfy different forms of Lyapunov stability (Justus, 2008), yielding increased empirical robustness. However, Huang et al. (2020) have shown that these empirical robustness improvements might be due to gradient obfuscation (Athalye et al., 2018) caused by the use of adaptive step-size solvers. Furthermore, Carrara et al. (2022) have shown that varying the solver tolerance during inference can increase empirical robustness.

**Verification and Reachability Analysis of NODEs**   Lopez et al. (2022) analyze the dynamics of very low dimensional ($d < 10$) NODEs using CORA (Althoff, 2013) and the (polynomial) Zonotope domain, and those of higher dimensional linear NODEs using the star set domain. In contrast to our work, they analyze only the learned dynamics, excluding the solver behavior, which has a significant effect on practical robustness (Huang et al., 2020). Grunbacher et al. (2021) introduce stochastic Lagrangian reachability to approximate the reachable sets of NODEs with high confidence by propagating concrete points sampled from the boundary of the input region. However, the number of required samples depends exponentially on the dimension of the problem, making it intractable for the high-dimensional setting we consider. Huang et al. (2022) propose forward invariance ODE, a sampling-based verification approach leveraging Lyapunov functions. Moreover, when using fixed step size ODE solvers the verification of NODEs can be seen as verifying neural network dynamic models (Adams et al., 2022; Wei & Liu, 2022) or by unrolling them even conventional feed-forward neural networks.

**Neural Network Verification**   Deterministic neural network verification methods, typically either translate the verification problem into a linear (Palma et al., 2021; Müller et al., 2022; Wang et al., 2021; Xu et al., 2021), mixed integer (Tjeng et al., 2019; Singh et al., 2019b), or semidefinite (Raghunathan et al., 2018; Dathathri et al., 2020) optimization problem, or propagate abstract elements through the network (Singh et al., 2019a; Gowal et al., 2019; Singh et al., 2018a) To obtain models amenable to certification, certified training (Mirman et al., 2018; Gowal et al., 2018; Zhang et al., 2020) methods use the latter class of approaches to compute and optimize a worst-case over-approximation of the training loss. However, none of these methods support the analysis of NODEs without substantial extensions.

## 8 CONCLUSION

In this work, we propose the analysis framework GAINS, **G**raph based **A**bstract **I**nterpretation for **NODE**s, which, for the first time, allows the verification and certified training of high dimensional NODEs based on the following key ideas: i) We introduce CAS solvers which retain the efficiency of adaptive solvers but are restricted to discrete instead of continuous step-sizes. ii) We leverage CAS solvers to construct efficient graph representations of all possible solver trajectories given an input region. iii) We build on linear bound propagation based neural network analysis and propose new algorithms to efficiently operate on these graph representations. Combined, these advances enable GAINS to analyze NODEs under consideration of solver effects in polynomial time.

## 9 ETHICS STATEMENT

As GAINS, for the first time, enables the certified training and verification of NODEs, it could help make real-world AI systems more robust to both malicious and random interference. Thus any positive and negative societal effects these systems have already could be amplified. Further, while we obtain formal robustness guarantees for $\ell_\infty$-norm bounded perturbations, this does not (necessarily) indicate sufficient robustness for safety-critical real-world applications, but could give practitioners a false sense of security.

## 10 REPRODUCIBILITY STATEMENT

We publish our code, all trained models, and detailed instructions on how to reproduce our results at `https://github.com/eth-sri/GAINS` and provide an anonymized version to the reviewers. Further algorithmic details can be found in App. A and B. Additionally, in App. C–E we provide implementation details and further discussions for our general method, classification tasks, and time-series forecasting tasks resistively. Lastly, details on the adversarial attacks and LCAP dataset used in §6.3 can be found App. F and G respectively.

## ACKNOWLEDGEMENTS

We would like to thank our anonymous reviewers for their constructive comments and insightful questions.

This work has been done as part of the EU grant ELSA (European Lighthouse on Secure and Safe AI, grant agreement no. 101070617) and the SERI grant SAFEAI (Certified Safe, Fair and Robust Artificial Intelligence, contract no. MB22.00088). Views and opinions expressed are however those of the authors only and do not necessarily reflect those of the European Union or European Commission. Neither the European Union nor the European Commission can be held responsible for them.

The work has received funding from the Swiss State Secretariat for Education, Research and Innovation (SERI).

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
