# OpenReview forum: "Efficient Certified Training and Robustness Verification of Neural ODEs"
_ICLR.cc/2023/Conference — ICLR 2023 poster_

### Official Review · Reviewer_CQh5 · 2022-10-19

**Confidence:** 4
**Correctness:** 4
**Technical Novelty And Significance:** 3
**Empirical Novelty And Significance:** 2
**Recommendation:** 6

**Clarity, Quality, Novelty And Reproducibility:**

See above.

Some minor notes / typos:
- The plots of Step size vs. Time (in Figure 1. or Figure 3.) allow the step size to go negative, is that possible / normal?
- The caption of Figure 7 is kind of overlapping with the legend of the figures.
- At the end of section 5, the authors discuss the fact that "eps-annealing alone is insufficient to stabilize the training". I did not see any discussion of annealing prior to that in the paper, but it is discussed in the appendix. Either some more of the discussion on training should be moved to the main paper, or this bit should be added to the appendix, because as it is, the context for it is not present.
- Second line of section 6.2 -> "We rescaling most features"
- Caption of Figure 9 -> "Comparison of th "



**Strength And Weaknesses:**

# Strengths
The paper proposes a clearly novel and interesting solution to be able to perform formal verification and bound propagation on a model for which existing methods would not work. The proposed solution is interesting, builds on existing verification research (DeepPoly) while clearly delineating their own contribution, and makes intuitive sense.
Relevant context (explanation about adversarial robustness and neural ODE) is given to make the paper accessible.

The author have provided their codebase as supplementary material, and have thoroughly detalied their experiments in the appendix, which makes me think that the work should be quite reproducible.

The result seems to be what you would expect with more traditional models: models trained without consideration for later verification are hard (or here apparently impossible) to verify, but incorporating the bounding procedure during the training makes it much easier to perform verification, at the cost of some nominal accuracy.


# Weakness
The part where I am actually a bit hesitant is on the applicability of this research. I'm mostly familiar with research on neural network verification, and less so on Neural ODE. Is there area where they particularly shine? Observing the experiments, what I see is that if I compare the adversarial and certified accuracy obtained on MNIST (83.9% for eps=0.1), they are significantly worse than simple convnets trained with IBP (99.77% for the same eps), in addition to being much more complex.
As a consequence, even if the results are technically interesting, I wonder if they are actually important.

I also note that the author make a claim that restricting step sizes to an exponential grid has "minimal impact on solver efficiency". I think that this should be evaluated beyond the results of Figure 4., which seems to be only for a single ODE. Would it be possible to perform this evaluation on more complex ODE, such as the one that might be defined by a Neural ODE?

## Question about training.
In Section 5., it is described that IBP bounds are used for training, and that trajectory needs to be sampled. Given the description of section 5., it seemed to me that it would be possible to actually compute a true upper bound. Is the decision to sample trajectories based on making the training iterations faster / use less memory, or is there another reason why it's not possible to use the true upper bound?



**Summary Of The Paper:**

This paper deals with the verification of Neural Ordinary Differential Equations (NODEs), which some papers have claimed to have more inherent robustness than standard neural networks.
Evaluating a NODE is based on solving the ODE defined by an initial condition (the input to the model) and a dynamics model, given by a trained neural network. Different types of ODE solvers might be used for the solution. This goal of this paper is to be able to obtain bounds on the outputs of the ODE, such that their robustness can be evaluated rigorously.

This work propose:
- to restrict the possible values that the step size of the solver can take. This allows to get some of the benefits of adaptive solvers, while maintaining the property that the solver can only go through a finite number of states (a state being determined by a time $t$ and a step size $h$). The fact that only a finite number of states can be reached makes it feasible to evaluate bounds on the network output (it just becomes dependent on computing bounds for each possible state)
- an algorithm to compute bounds on such a computational graph. Usually, bound computation is done on fixed computational graphs, where you can define exactly the function of each node (broadly, a node in the computational graph to bound is an activation of a network and can only have been produced in a given way). With the proposed formalism, each node (a given state) may have been reached by doing a variable number of steps of the solver, so to obtain a bound, you need to pick the worst case bound across all the possible paths that may have been taken. The author explain how to do this both for simple IBP bounds, as well as for the more complex linear bounds.

Evaluation is performed on image classification, using MNIST and FashionMNIST, as well as on time series regression.

**Summary Of The Review:**

The paper is rigorous and provide a good solution to the question "how can you propagate bounds through the solving of a Neural ODE". The method discussed is presented well. The main doubt is whether that question is worth solving in the first place.


[Note]
I trust myself to review the elements of the paper that relate to verification but don't have significant expertise in Neural ODE.

---

> ### Author Response · Authors · 2022-11-13
> **Response to Reviewer CQh5**
>
> $\newcommand{Rf}{\textcolor{orange}{CQh5}}$
> We thank reviewer $\Rf$ for the in-depth review, raising interesting questions, and making insightful suggestions. We are encouraged that they recognize the novelty and quality of our solution as well as the clarity of its representation. Below we answer their remaining questions.
>
> **Q: Can you discuss the relevance of NODEs and in particular their verification?**
> Yes. NODEs are an active research area, with recent results including remarkable performance in few-shot learning [1], motion control [2], and time-series forecasting [3]. While the first of these is only just being explored, the latter are an obvious fit with NODEs, due to their ability to inherently model dynamic processes.
> In motion control, theoretically-derived dynamics can be augmented using a NODE to correct modeling errors in complex systems such as drones and industrial robots, where verification is essential to certify safety properties.
> In time-series forecasting, NODEs are uniquely suited to handle in-homogenous time series, as shown in our Physio-Net experiments.
> While we agree that NODEs are not an ideal fit for computer vision tasks, they are a popular benchmark for both NODEs and neural network verification making them an ideal proxy task for us.
>
> **Q: Can you discuss the claim that restricting step sizes to an exponential grid has only a "minimal impact on solver efficiency"?**
> We make this claim based on our theoretical insight that (under mild assumptions) the step sizes chosen by a CAS solver will at most be a factor of $\alpha$ (2 for our experiments) smaller than those of AS leading to asymptotically identical performance. We have added more experiments (see Appendix C.2 and H.1) confirming these results and showing that CAS can even outperform AS solvers in terms of both precision and speed. For more details, please see the main response.
>
> **Q: Why are trajectories sampled during training instead of computing sound bounds?**
> As the reviewer suggests, trajectories are sampled, leading to approximate instead of sound bounds, in order to reduce memory and time complexity, allowing larger batch sizes to be used during training.
>
> **Minor points:**
> * The step size origin in Figures 1 and 3 is not 0, but the initial step size $h_0$, we have moved the origin to avoid confusion.
> * We updated Figure 7 to avoid caption and legend overlapping.
> * We have moved the paragraph “Stabilizing Training” to Appendix B and provide the necessary context there.
> * We fixed the typos. Thank you for pointing them out!
>
> We hope to have been able to address the reviewer's comments, are happy to answer any follow-up questions, and are looking forward to their reply.
>
> **References**
> [1] Zhang et al. “MetaNODE: Prototype Optimization as a Neural ODE for Few-Shot Learning”, AAAI 2022
> [2] Vantilborgh et al. “Efficient ODE Substructure Identification of the Acrobot under Partial Observability using Neural Networks and Direct Multiple Shooting”, AIM 2022
> [3] Zhou et al. “Forecasting Reservoir Inflow via Recurrent Neural ODEs”, AAAI 2021

---

### Official Review · Reviewer_LBBD · 2022-10-24

**Confidence:** 3
**Correctness:** 3
**Technical Novelty And Significance:** 3
**Empirical Novelty And Significance:** 3
**Recommendation:** 8

**Clarity, Quality, Novelty And Reproducibility:**

The authors are the first to propose an analysis framework for verification of high dimensional NODEs. The paper is well written, the context and problem formulation are introduced clearly and the most important aspect of contribution are explained well. The theoretical results given seem to be theoretically sound. The theoretical and experimental details are clearly documented, and the code for implementation of the work is shared.

**Strength And Weaknesses:**

One of the main contributions of this paper is the reduction of the computational complexity of verification of NODES by GAINS enabled by controlled adaptive ODE solvers. The theoretical impact of the use of a CAS over an AS solver has been briefly and clearly described in 'Comparison to Adaptive Solvers' on page 5. However, the included empirical comparison is difficult to grasp. Moreover, from Figure 4, it appears that for a small error threshold, the difference in performance between CAS and AS is relatively large. It would be good if the authors could include for some of the experiments on the (F)MNIST and PHYSIO-Net datasets the difference in performance and (standard) adversarial robustness of the AS and CAS solvers. Furthermore, the authors should clarify the impact of the update factor on the computational complexity.

The authors perform an extensive empirical evaluation of the performance of the framework. These experiments clearly show the strength of the proposed framework to assess and improve adversarial robustness. It would be interesting to compare the performance of the networks trained by GAINS and TisODE or other approaches as cited in the related work that improve empirical robustness.

Furthermore, it should be emphasised in the related works that for an ODE solver with fixed time step, the problem considered in this work is similar to that of verifying neural networks dynamic models, which are  discrete time dynamical models driven by a neural network for which formal robustness verification and control against have been recently studied [1,2].

[1] : Adams, Steven, et al "Formal control synthesis for stochastic neural network dynamic models." IEEE Control Systems Letters (2022).

[2] :  Wei, Tianhao, and Changliu Liu. "Safe Control with Neural Network Dynamic Models." Learning for Dynamics and Control Conference. PMLR, 2022.



**Summary Of The Paper:**

This paper proposes a framework based on linear bound propagation that takes advantage of a new class of ODE solvers to enable training and verification of Neural Ordinary Differential Equations. Specifically, the proposed new class of adaptive ODE solvers, CAS, is based on variable but discrete time steps such that the solver trajectories can be captured in a graph representation. This reduces the runtime of NODES from intractable exponential computation time to polynomial time.

**Summary Of The Review:**

To the best of the my knowledge, this work is the first to present a scalable verification framework for NODEs. The paper is well written, and the theoretical contribution seems sound. The authors included an extensive empirical evaluation of the performance of the framework. The authors should explain in more detail the restrictiveness of the proposed controlled adaptive ODE solver.

---

> ### Author Response · Authors · 2022-11-13
> **Response to Reviewer LBBD**
>
> $\newcommand{Rtr}{\textcolor{blue}{LBBD}}$
> We thank reviewer $\Rtr$ for their insightful questions and helpful suggestions. We are encouraged that they recognize the novelty of our approach, our extensive evaluation, and the clarity of presentation. Below we answer the remaining questions.
>
> **Q: Can you extend the (empirical) comparison between CAS and AS?**
> We have added experiments on multiple NODEs demonstrating that CAS and AS perform very similarly (Appendix C.2 and H.1). For a more detailed answer, we refer the reviewer to the main response.
>
> **Q: Can you clarify the impact of the update factor on the computational complexity?**
> Yes. Including the update factor $\alpha$ as a variable instead of a constant in the derivation, we obtain a complexity of $\mathcal{O}(T_{end}^2 log^2(T_{end})log^{-2}(\alpha))$. We have added a detailed derivation also considering the minimum step size $h_\text{min}$ in Appendix B.
>
> **Q: Can you compare the performance GAINS trained NODES with TisODEs?**
> Yes. We added an experiment comparing GAINS-trained NODES with TisODEs on MNIST. We observe that while the TisODE has better standard accuracy, its adversarial accuracy quickly decreases to only $55.5\%$ at $\epsilon=0.2$, where the GAINS-trained NODE still has $84.5\%$ adversarial accuracy. Further, we highlight that TisODEs are not trained with later verification in mind, explaining the gap in standard accuracy. For more details, see Appendix H.2 and Table 8.
>
> **Q: Can you discuss the verification of fixed step size ODE solvers in the related work?**
> As pointed out by the reviewer, the verification of NODEs with fixed step size solvers reduces to the verification of neural network dynamic models or, via unrolling these, even to standard neural network verification. We have updated related work accordingly including the suggested references. However, we want to point out that fixed step size solvers are typically not used for NODEs.
>
> We hope to have addressed all of the reviewer's comments, are happy to answer any follow-up questions, and are looking forward to the reviewer’s response.

---

> > ### Comment · Reviewer_LBBD · 2022-11-18
> > **Thanks for your response**
> >
> > I thank the authors for their reply. They have addressed my comments, consequently I maintain my positive score.

---

### Official Review · Reviewer_yJLc · 2022-10-28

**Confidence:** 4
**Correctness:** 4
**Technical Novelty And Significance:** 3
**Empirical Novelty And Significance:** Not applicable
**Recommendation:** 8

**Clarity, Quality, Novelty And Reproducibility:**

The presentation and clarity are very good.

The novelty and quality of technical contribution is also good.

The code is provided for reproducibility.

**Strength And Weaknesses:**

### STRENGTHS
1. Novel Problem:
The verification of neural ODEs is an under-explored area.
2. Novel solution:
I like that the solution is built on top of a propagation based method, which makes it versatile and easier to develop in the future. Furthermore, it is clear that the authors had to do sufficient work to first make CAS solvers, then
3. Good writing:
The paper is very well written. The figures are also very helpful for understanding the method.

### WEAKNESSES
1. Validate CAS vs AS on more problems
- Could you please add experiments on more common problems to compare the two solvers. It is important to make sure that CAS is not much inferior to usual AS solvers.
2. Certified accuracy of even adversarial trained methods is 0 everywhere
- This is quite surprising. It would be important to understand why this is happening.  Does this mean that the bounds produced by the method are very weak? I would recommend the following experiments
- Could you plot a figure similar to Fig.2 from Beta-CROWN 9 (https://arxiv.org/pdf/2103.06624.pdf) as this will help us understand the tightness of the bounds.
- Can you reduce the epsilon further to see if we can verify adv trained networks at all?
3. Medium scale experiments missing
- It would be good to have experiments on CIFAR-10. It is common to use even Imagenet-32 in Neural ODE papers, so running on CIFAR-10 should not be a trouble. Point 2 above scares me that your method might not scale well scale well.
- Baseline: Would it be possible to add a baseline into the tables? Maybe an inferior version of your own method by removing some component?



**Summary Of The Paper:**

### Problem
The paper tackled robustness certification and certified training of Neural ODEs.

### Proposed Method
Authors claim that directly verifying adaptive solvers is difficult. So authors propose a modified solver called controlled adaptive ODE solver. This essentially restricts the step-sizes to a discrete set, which in turn restricts states that the solver can reach compared to the usual adaptive solvers.

The authors then propose a novel verification method for this solver, which is built on top of propagation-based verification methods like DeepPoly or CROWN.

Authors also show how this method can be used for certified training.

### Experiments
Authors conduct two sets of experiments, on classification on MNIST and FMNIST, and time-series on Physio-net.

**Summary Of The Review:**

The problem and proposed method are novel. But I am concerned about the scalability of the technique. The bounds don't seem to be very tight on MNIST. Experiments missing on a medium scale dataset like CIFAR-10 make it difficult to validate this. I am on the borderline (5/6) for this reason at the moment and willing to change my rating depending on responses to my concerns.

---

> ### Author Response · Authors · 2022-11-13
> **Response to Reviewer yJLc**
>
> $\newcommand{Rt}{\textcolor{green}{yJLc}}$
>
> We thank reviewer $\Rt$ for the detailed review, interesting questions, and insightful comments. We are particularly encouraged that the reviewer appreciates the novelty of our approach and the clarity of its presentation. Below we address their remaining comments.
>
> **Q: Can you compare the performance of CAS and AS solvers on more problems?**
> We have added experiments on multiple NODEs demonstrating that CAS and AS perform very similarly (Appendix C.2 and H.1), for a more detailed answer, we refer the reviewer to the main response.
>
> **Q: Can you discuss the tightness of the obtained bounds?**
> To evaluate the tightness of our certified bounds, we compare them to empirical ones, obtained using a strong adversarial attack (similar to the referenced Fig. 2 from $\beta$-CROWN [1]) on a GAINS-trained NODE for Fashion-MNIST. We illustrate the results both in a scatter plot (Figure 14 left) and in a histogram showing the frequency of different bound tightnesses (Figure 14 right). We observe that GAINS mostly yields tight lower bounds which lie only marginally below the empirical upper bounds. We invite the reviewer to look at Figure 14 in Appendix H.3 for more detail.
>
> **Q: Can you discuss why the certified accuracy of even adversarially trained NODEs is zero?**
> For NODEs that were not trained with later verification in mind, we indeed only obtain vacuous bounds. This is mostly due to the combination of two effects: (i) Box abstractions grow exponentially with model depth unless a network is specifically regularized and trained to avoid this blow-up [2]. (ii) NODEs have extremely large effective depths. While the adversarially trained networks analyzed by Wang et al. [1] in Figure 2 have a depth of only 3 ReLU layers, the large number of timesteps and solver calls per time-step leads to effective depths of up to over 60 layers for NODEs in the image classification and 100s for PhysioNet. Consequently, special training is required to avoid this blow-up and make an analysis with GAINS effective.
> We point out that $\beta$-CROWN [1] is a complete verifier, i.e. given enough (exponential) time, it can decide any property for standard feed-forward neural networks. This is not the case for GAINS, which is a so-called incomplete verifier. For $\beta$-CROWN the main challenge was to increase analysis precision while building on other successful verification approaches. For NODEs, no such successful (scalable) prior work exists, thus the challenge here is to enable verification in the first place and thereby lay a stepping stone for future work.
>
> **Q: Can you discuss the scalability of the proposed method?**
> Please see the detailed answer in the main response.
>
> **Q: Can you compare your approach to a baseline?**
> Excellent suggestion! We added an ablation study to Appendix H.3 where we compare GAINS to interval bound propagation (still using our trajectory graph construction), which we call GAINS-BOX.
> Comparing the resulting certified accuracies on Fashion-MNIST, we observe that GAINS is more precise in all settings, obtaining up to $12$ percentage points higher certified accuracies (see Table 9).
> As a more fine-grained measure of precision, we compare our certified bounds on the worst-case logit differences to empirical ones, obtained using a strong adversarial attack (similar to the referenced Fig. 2 from $\beta$-CROWN [1]). Illustrating the resulting bound tightness (difference between empirical and verified) bound for a GAINS-trained network both as in  $\beta$-CROWN [1] and showing their frequency in a histogram (see Figure 14), we observe that GAINS yields significantly tighter bounds than GAINS-BOX, demonstrating the importance of linear bound propagation and thus CURLS.
> Upon publication, we will use the extra space awarded by the camera-ready version to move these results forward to the main paper.
>
> We hope to have been able to address the reviewers’ concerns, are happy to answer any follow-up questions, and are looking forward to their reply.
>
> **References**
> [1] Wang, et al. "Beta-crown: Efficient bound propagation with per-neuron split constraints for neural network robustness verification." NeurIPS 2021
> [2] Shi, et al. "Fast certified robust training with short warmup." NeurIPS 2021

---

> > ### Comment · Reviewer_yJLc · 2022-11-17
> > **Thanks for your response**
> >
> > I would like to thank the authors for their reply.
> > The authors have addressed all my concerns in their replies and the supplementary paper so I have updated my rating, thanks.

---

### Official Review · Reviewer_bH4g · 2022-11-02

**Confidence:** 3
**Correctness:** 4
**Technical Novelty And Significance:** 3
**Empirical Novelty And Significance:** 3
**Recommendation:** 6

**Clarity, Quality, Novelty And Reproducibility:**

The paper is well written and includes some solid results on certified robustness of the neural ODEs. Codes have been provided to reproduce the results. I think the idea of combining ODE solvers with variable steps and the graph representation of solver trajectories is interesting. I am a little bit concerned with the scalability of the proposed approach, since MNIST is really considered as a very simple task.

**Strength And Weaknesses:**

Strength:
1. The combination of ODE solvers with variable steps and the graph representation of solver trajectories seem to be interesting.

2. The improvement of running time is significant.

Weaknesses:

1. MNIST is a very simple task. Is it possible to run the proposed method for CIFAR10?

2. The theoretical novelty of this paper is unclear.

**Summary Of The Paper:**

This paper proposes a method called GAINS to address the robustness certification of neural ODEs via combining ODE solvers with variable steps and an efficient graph representation of solver trajectories. The authors provide some arguments showing the proposed approach significantly reduce the run time.  Numerical study is also performed on MNIST, FMNIST, and Physio-Net to demonstrate the certified robustness of the proposed method.

**Summary Of The Review:**

For now I give a "6." Depending on how the authors address my concerns (on scalability and theoretical novelty), I may either increase or decrease my score.

---

> ### Author Response · Authors · 2022-11-13
> **Response to Reviewer bH4g**
>
> $\newcommand{Ro}{\textcolor{purple}{bH4g}}$
> $\newcommand{Rt}{\textcolor{green}{yJLc}}$
> $\newcommand{Rtr}{\textcolor{blue}{LBBD}}$
> $\newcommand{Rf}{\textcolor{orange}{CQh5}}$
> We thank reviewer $\Ro$ for raising interesting questions and making helpful suggestions. We are encouraged that they find our work interesting, significant, and well-written. Below we answer their remaining questions.
>
> **Q: Can you discuss the scalability of the proposed approach (e.g. to CIFAR10)?**
> We added new results on the CIFAR10 dataset (see Table.10 in Appendix H.4) and refer the reviewer to the main response for a detailed answer.
>
> **Q: Can you clarify the theoretical novelty of the proposed approach?**
> Yes. We are the first to propose a (comparatively) scalable approach for the verification of NODEs and the first to consider solver effects as part of any NODE verification. To this end, we introduce a novel solver class (CAS), trajectory representation (trajectory graph), and  algorithm to operate on this representation (CURLS). We combine these advances in the verification framework GAINS. We believe this constitutes significant novelty, as recognized by reviewers $\Rt$, $\Rtr$, and $\Rf$.
>
> We hope to have been able to address the reviewer’s comments, are happy to answer any follow-up questions, and are looking forward to their reply.

---

> > ### Author Response · Authors · 2022-12-04
> > **Response Follow-Up**
> >
> > $\newcommand{Ro}{\textcolor{purple}{bH4g}}$
> > We again want to thank the reviewer $\Ro$ for their insightful review.
> > As the discussion period is coming to a close, we would greatly appreciate feedback on whether we were able to address the reviewer's comments and are happy to answer any follow-up questions.

---

### Author Response · Authors · 2022-11-13
**Main Response**

$\newcommand{Ro}{\textcolor{purple}{bH4g}}$
$\newcommand{Rt}{\textcolor{green}{yJLc}}$
$\newcommand{Rtr}{\textcolor{blue}{LBBD}}$
$\newcommand{Rf}{\textcolor{orange}{CQh5}}$
We thank the reviewers for their valuable comments, insightful questions, and helpful suggestions. We are particularly encouraged to hear that they appreciate the novelty ($\Rt$, $\Rtr$, $\Rf$) and quality ($\Rt$, $\Rtr$, $\Rf$) of our work as well as the clarity of its presentation ($\Ro$, $\Rt$, $\Rtr$, $\Rf$). We have updated our paper (see blue sections) to incorporate the reviewers’ feedback and identified two common questions which we answer below, addressing reviewer-individual points in separate replies.

**Q: Can you discuss the scalability of the proposed approach? ($\Ro$, $\Rt$)**
First, we point out that compared to the only prior work handling deterministic NODE verification [1] (although ignoring solver effects) we scale to over 50-times larger latent spaces, demonstrating the significant improvement GAINS makes over the state-of-the-art.
Second, we highlight that combining the different measurement times of the PhysioNet dataset, NODEs can handle inputs with up to 10-times larger dimensionality (29k) than CIFAR10. We have added an experiment on CIFAR10 (see Table.10 in Appendix H.4) where we reach 57% certified accuracy.
Third, we highlight that verification methods typically scale only to comparatively small networks [2], with state-of-the-art performance in computer vision tasks typically being reached with only 7-layer CNNs [3]. Scaling to datasets beyond CIFAR10 remains an open problem, e.g, the state-of-the-art for TinyImageNet is below 20% certified accuracy at less than 30% natural accuracy [3].
We thus believe that GAINS is a promising step in NODE verification that future work can build on.

**Q: Can you extend the comparison between CAS and AS solver? ($\Rt$, $\Rtr$, $\Rf$)**
Yes. We perform two experiments. The first compares absolute errors depending on the number of performed steps on a range of NODEs, the second compares the resulting standard and adversarial accuracies.
To compare absolute errors over the number of solver steps in a more challenging setting than the ODE in Figure 4, we train NODEs on MNIST and Fashion-MNIST using an AS and report results on the first 1000 test-set samples, computing a reference solution with a 100-times smaller error threshold. We observe (numbers for Fashion-MNIST) that while CAS solvers tend to perform slightly more steps (13.5 vs 12.2 on average), they obtain notably smaller absolute errors both for the same number of steps (ratio between mean absolute errors of 2.76) and overall (ratio of 2.01). If we reduce the error threshold of the AS by a factor of 2, the CAS solver performs fewer steps on average (13.5 vs 14.7) and is still more precise (error ratio of 1.35). We believe this to be due to the more conservative step-size updates of CAS solvers, which could become a disadvantage if the initial step size is chosen very poorly. For NODEs, however, CAS seems to be very competitive with AS solvers. For a more detailed description of these results, please see Appendix C.2 and Figure 13.
To compare standard and adversarial accuracy obtained with CAS and AS solvers, we train and evaluate NODES using either method for adversarial and standard training on both MNIST and Fashion-MNIST. Reporting the mean and standard deviation across 3 runs,  we observe small differences in mean performance, but not a single setting where the $\pm 1$ standard deviation intervals do not overlap. We thus conclude that any performance differences between CAS and AS solvers are negligible for NODEs compared to the effect of certified training. For a more detailed description of these results, please see Appendix H.1 and Table 7.
Upon publication, we will use the extra space awarded by the camera-ready version to move these results forward to the main paper.

We hope to have been able to address the reviewers’ questions and concerns and are happy to answer any follow-up questions.

**Resources**
[1] Lopez et al. "Reachability analysis of a general class of neural ordinary differential equations." arXiv 2022
[2] Bak et al. "The second international verification of neural networks competition (VNN-Comp 2021): Summary and results." arXiv 2021
[3] Shi, et al. "Fast certified robust training via better initialization and shorter warmup." NeurIPS 2021

---

### Author Response · Authors · 2022-11-17
**Response Follow-Up**

We thank the reviewers again for their high-quality reviews and want to remind them that PDF updates will only be possible for two more days. Therefore, we would greatly appreciate it if they could let us know, before that deadline, if they would like any further clarifications or have any follow-up questions.

---

### Decision · Program_Chairs · 2023-01-20

**Decision:**

Accept: poster

**Justification For Why Not Higher Score:**

The authors do not show scalability of the method to complex tasks where Neural ODEs would have a significant advantage.

**Justification For Why Not Lower Score:**

The authors develop efficient and novel certification and certified training algorithms for a challenging class of neural models, i.e., neural ODEs.

**Metareview: Summary, Strengths And Weaknesses:**

The authors develop a novel certified robustness training and verification method for neural ODEs. By restricting the step-sizes taken in a neural ODE solver, they are able to reduce the problem of certifying robustness of neural ODEs to that of standard computation graphs, enabling them to take advantage of efficient certification methods like CROWN and DeepPoly.

Strenghts:
1. The paper achieves SOTA certified robustness results for Neural ODEs.
2. The authors obtain an exponential speedup relative to prior work on Neural ODE certification.

Weaknesses:
1. The paper only presents experiments on smaller datasets and tasks, calling into question the scalability of the method to larger datasets and complex prediction tasks where Neural ODEs are likely to be used.

In summary, the reviewers were satisfied with the contributions of the paper and have consensus on acceptance of the paper.

**Note From Pc:**

if the above contains the word "oral" or "spotlight" please see: "oral" presentation means -> notable-top-5% and "spotlight" means -> notable-top-25%. As stated in our emails, we are disassociating presentation type from AC recommendations

**Summary Of Ac-Reviewer Meeting:**

No meeting